# Dynamics of Florida milk production and total phosphate in Lake Okeechobee

**Joseph Park**[1,2]*, **Erik Saberski**[3], **Erik Stabenau**[1], **George Sugihara**[3]

**1** U.S. Department of the Interior, South Florida Natural Resources Center, Homestead, FL, United States of America, **2** Department of Engineering and Development, United Nations Comprehensive Nuclear-Test-Ban Treaty Organization, Vienna, Austria, **3** Scripps Institution of Oceanography, University of California San Diego, La Jolla, CA, United States of America

* JosephPark@IEEE.org

**Data Availability Statement:** The minimal data set is available on Zenodo: http://doi.org/10.5281/zenodo.5034959.

**Funding:** This work was funded in collaboration of the U.S. Department of the Interior, National Park Service, Everglades National Park, and University

## Abstract

A central tenant of the Comprehensive Everglades Restoration Plan (CERP) is nutrient reduction to levels supportive of ecosystem health. A particular focus is phosphorus. We examine links between agricultural production and phosphorus concentration in the Everglades headwaters: Kissimmee River basin and Lake Okeechobee, considered an important source of water for restoration efforts. Over a span of 47 years we find strong correspondence between milk production in Florida and total phosphate in the lake, and, over the last decade, evidence that phosphorus concentrations in the lake water column may have initiated a long-anticipated decline.

## Introduction

### Historical perspective

Prior to the 19[th] Century, the Florida Everglades consisted of 3 million acres of marsh draining the Kissimmee River Basin and Lake Okeechobee southward into Florida Bay. Water flowing into Lake Okeechobee came primarily from the Kissimmee River, meandering approximately 103 miles as a 1 to 2 mile-wide floodplain. The shallow, slow moving flow provided conditions well-suited for nutrient uptake, contributing to low nutrient concentrations throughout the system. As a result, addition of even small amounts of nutrients can significantly effect the structure and productivity of the native ecosystem [1].

Consistent with ideals of *manifest destiny*, efforts to "drain" the Everglades to produce arable lands were initiated in the late 19[th] Century, and, in the 1950's, the Kissimmee Flood Control project replaced the original meandering geometry with a channel consisting of straight-line segments [2, 3]. Completion of the project coincided with increased phosphorus loads to Lake Okeechobee from the transport of phosphorus-laden sediments [4, 5]. A comparative rendition of the pre-development and current systems is shown in Fig 1.

The extensive spread of agriculture in the upstream drainage basins also contributed to this increased load. Phosphorus is added to uplands in fertilizers, organic solids (e.g., animal wastes, composts, crop residues), wastewater, and animal feeds. Some phosphorus is exported from the drainage basin as agricultural products, however, a significant amount accumulates

of California San Diego through the Cooperative Ecosystem Studies Units (CESU) Network http://www.cesu.psu.edu/. This work was also supported by the DoD-Strategic Environmental Research and Development Program 15 RC-2509, NSF DEB-1655203, NSF ABI-1667584, DOI USDI-NPS P20AC00527, NSF-IOS 1936674, the Scripps Institution of Oceanography Postdoctoral Fellowship, the McQuown Fund, and, the McQuown Chair in Natural Sciences, University of California, San Diego. The funders had no role in study design, data collection and analysis, decision to publish, or preparation of the manuscript.

**Competing interests:** The authors have declared that no competing interests exist.

in upland soils and sediments, and, a portion is then transported by surface flow to the lake [6].

Historically, cattle ranching was the main agricultural use of the watershed north of the lake, however, in the 1950s dairy farming increased eight-fold, with a corresponding increase in phosphorus exports from 250 to 2,000 metric tons per year [7]. In 2000, Florida enacted the Lake Okeechobee Protection Act (Chapter 00–103, Laws of Florida), mandating a comprehensive plan to reduce watershed phosphorus loading to meet a total maximum daily load (TMDL) of 105 metric tons (mt) per year of surface-water input by 2015.

## Contemporary conditions & restoration

Over the last two decades, many sources have been remediated, producing significant declines in source loadings [7–15]. On-site monitoring at the farm level has demonstrated improvement, particularly for intensive land uses such as dairies where treatment systems, stormwater

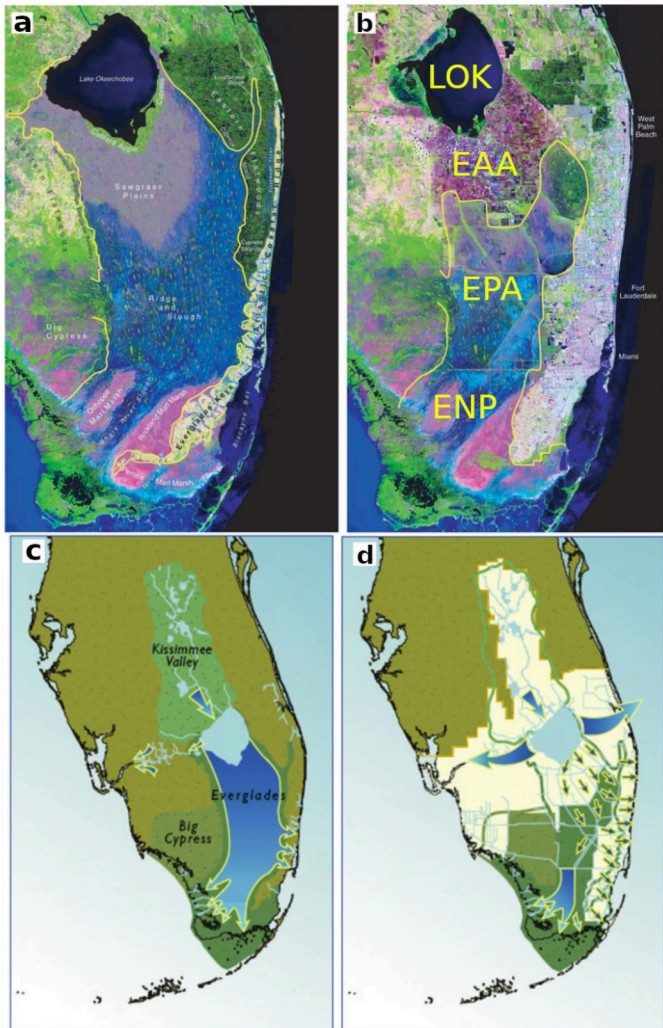

**Fig 1. South Florida satellite image.** a) Simulated South Florida satellite image circa 1850. b) Satellite image 1994. c) Pre-development hydrologic flow paths. d) Current flow paths. LOK—Lake Okeechobee. EAA—Everglades Agricultural Area. EPA—Everglades Protection Area. ENP—Everglades National Park.

management, and reuse have been implemented [16]. However, a large amount of legacy phosphorus remains throughout the watershed [17], and, owing to the significant variability in phosphorus dynamics, it is not clear that total phosphorus concentrations have significantly declined [18].

A longstanding presumption of CERP is that substantial quantities of "new water" will be delivered from Lake Okeechobee, accordingly, water quality in the lake has important implications on Everglades restoration [18]. However, currently, Lake Okeechobee water has a limited effect on the downstream Everglades Protection Area (EPA) as roughly 4 percent of the lake's average outflow reaches the EPAs. The primary contributors are nutrients discharged from the Everglades Agricultural Area (EAA) and C-139 basins [3]. To address this, best management practices (BMPs) and stormwater treatment areas (STAs) have been designed and implemented to reduce phosphorus from the EAA basins into the Everglades Protection Area [19].

Specific to Lake Okeechobee, increased phosphorus concentrations alter the structure and functioning of the lake and downstream ecosystems. The chronic and substantial phosphorus increase has resulted in conversion of a phosphorus-limited system into a nitrogen-limited system. Attendent changes in the lake include increased algal blooms and abundance of nitrogen-fixing cyanobacteria [20]. For example, during summer 2016, a large bloom of the cyanobacterium *Microcystis aeruginosa* occurred in Lake Okeechobee, and, subsequently in the St. Lucie Estuary. These events were attributed to high nutrient levels supporting the growth of phytoplankton [18].

## Dynamical perspective

Conventional views find that total phosphorus concentrations in the lake have not significantly declined over the 1974–2017 period of record, despite the array of projects that have reduced phosphorous sources [12, 18]. However, there is significant variability across multiple timescales in the phosphorus data, with the potential to confound linear, block approaches of statistical interpretation. Here, we use two data-driven, nonlinear dynamical tools to examine time and cross variable dependence: Empirical Mode Decomposition (EMD) [21], and, Empirical Dynamic Modeling (EDM) [22].

EMD decomposes signals into scale-dependent modes termed intrinsic mode functions (IMF) without constraints of linearity or stationarity as presumed by Fourier, wavelet or Eigen decomposition. IMFs capture oscillatory modes, and, EMD residuals the nonlinear trends. Application of the Hilbert transform to IMFs provides time-dependent instantaneous frequency estimates, with the combination of EMD and IMF Hilbert spectra constituting the Hilbert-Huang transform (HHT). The Hilbert-Huang transform was motivated by the need for data-adaptive signal decomposition rather than one based on *a-priori* presumed basis, *ala* Fourier and wavelet decompositions. As many real-world systems express nonlinear, nonstationary dynamics, it has been found that IMFs are particularly astute at isolating physically-meaningful dynamics. Examples include structural damage detection, seismology, speech recognition, biological and geophysical signals, and, financial time series [23]. A comprehensive introduction and review of the HHT is found in reference [21].

EDM is a toolset to predict, explore, and, identify relationships of nonlinear dynamical systems. Nonlinear systems are state-dependent systems: states are determined by previous states wherein a specific set or sequence of states govern transition from one state to another. EDM operates in this space, the multidimensional state-space of system dynamics rather than on single dimensional observational time series. Further, EDM does not presume relationships among states, for example, a functional dependence, but predicts future states based on projections from localised, neighboring states. EDM is thus a state-space, nearest-neighbors

paradigm where system dynamics are inferred from states derived from observational time series. This provides a model-agnostic representation of the system naturally encompassing nonlinear dynamics without parametric presumptions, fitting statistics, or, specifying equations [24]. An accessible and complete overview of EDM is provided by Chang et. al. [22].

We examine the data with EMD and EDM to reveal underlying dynamics and relationships between milk production and lake phosphorus. The synthesis of EMD and EDM has been termed empirical mode modeling (EMM), where EMD IMF's are used to create physically relevant multivariable state spaces for EDM [25].

## Materials and methods

### Data

Milk production data are obtained from the United States Department of Agriculture (USDA) National Agricultural Statistics Service (NASS) database, reporting total monthly milk production in Florida from January 1970 through June 2020 [26]. Data are shown in Fig 2a.

Phosphorus data for Lake Okeechobee is a 5 station average (stations L001, L003, L004, L007, L008) obtained from the South Florida Water Management District (SFWMD) DBHydro environmental database [27]. Raw data span the period December 11, 1972 to August 8, 2020. Stations were selected based on criteria of a minimal 40 year period-of-record, and, subjected to manual quality assurance inspection.

The station average phosphorus time series is interpolated with a spline to monthly dates of the milk production data. The result is a data block of monthly milk production and interpolated total phosphate from January 1973 through June 2020 (Fig 2a and 2b).

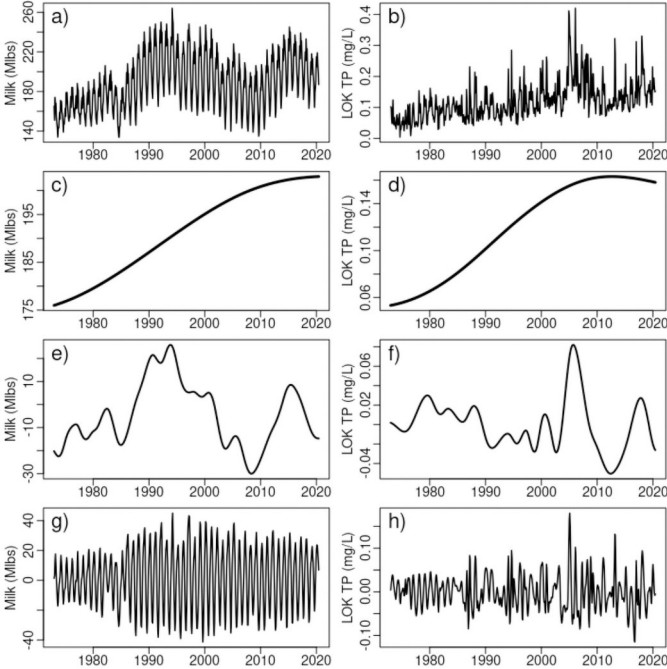

**Fig 2. Data and empirical mode decompositions.** a) Milk production b) LOK total phosphate c) Milk decadal d) Total phosphate decadal e) Milk interannual f) Total phosphate interannual g) Milk intra-annual h) Total phosphate intra-annual.

Data representing the El-Niño Southern Oscillation are monthly values of the Multivariate ENSO Index (MEI) obtained from the National Oceanic and Atmospheric Administration Physical Sciences Laboratory [28].

## Empirical mode modeling

We use EMD [29] to decompose the time series into IMFs and associated nonlinear residuals (see Fig 5 in S1 Appendix Empirical mode decomposition). We use the mean value of the instantaneous frequency of each IMF to represent the overall frequency, and thus inverse period, of the IMF. Intra-annual IMFs are deemed to have mean periods between 0.5 and 2 years. Interannual IMFs have periods greater than 3 years. Interannual time series are created by summation of IMFs with interannual frequencies, for milk production IMFs 4,5,6, and IMFs 5,6,7 for phosphorus. Intra-annual time series consist of IMFs 2 and 3 for milk production, and, IMFs 2,3,4 of phosphorus.

We then use raw data and IMFs of interannual and intra-annual modes in an EDM convergent cross mapping (CCM) analysis [30]. CCM identifies potential causal links between state variables based on information shared between multidimensional embeddings of the variables [31]. CCM can be viewed as a dynamically-informed, fully nonlinear, analog to cross correlation. However, instead of reliance on temporal or cross variable *coincidence*, CCM is based on affine mappings of dynamical system states where CCM values indicate the cross variable predictability. Convergence of predictability as the information content and density of the state-space increase indicate shared dynamics and a measure of causality [31].

Since the data are autocorrelated (lag-1 correlations of 0.86 and 0.70 for milk and phosphorus respectively), and, exhibit seasonal dynamics, we use an exclusion radius of 12 points (months). That is, in the EDM state-space nearest neighbor search for the prediction at each time step, neighbors that are temporally within the exclusion radius of 12 points (months) are excluded from the prediction. This prevents any influence of autocorrelation or seasonality on the CCM information assessment.

To assess significance of the CCM results, we employ surrogate data samples created from the random phase method of Ebisuzaki [32]. We use N = 1000 surrogate time series of the milk or phosphorus data, and compute an EDM cross map strength for each surrogate and the original time series. For example, the original phosphorus time series is EDM cross mapped against 1000 surrogate milk time series created from Ebisuzaki spectral phase randomisation.

Given the cross map strength of variables $X$ and $Y$, $\rho_{XY}$, and, a vector of cross map strengths $\rho_{XY_N}$ between $X$ and N surrogates $Y_N$, a p-value representing the probability of rejecting the null hypothesis that the cross map strength $\rho_{XY}$ is not due to randomness, can be specified as:

$$p = 1 - CDF(\rho_{XY_N})\big|_{\rho_{XY}} \tag{1}$$

where $CDF(\rho_{XY_N})$ is the distribution function of the surrogate cross map strengths.

## Results

One of the most striking results can be seen in a relative comparison of milk production and lake phosphorus at different time scales. Fig 3 presents scaled (mean offset, standard deviation normalized) comparisons of the raw data, nonlinear trends, interannual, and, intra-annual modes. In the nonlinear trends of panel b) there is a remarkable coherence on the decadal time scale with relative changes in milk production and lake phosphorus nearly identical from the mid 1970's through the early 2000's. In the final decade, 2010–2020, there is flattening of overall milk production, and, a noticeable inflection towards phosphorus reduction. This may

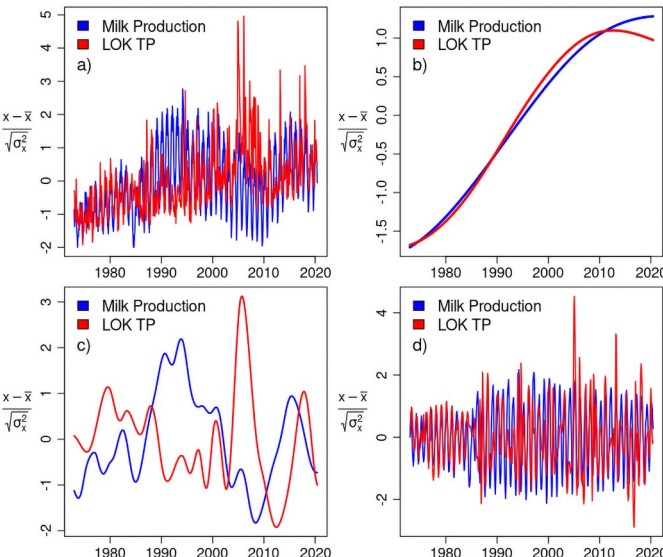

**Fig 3. Empirical mode decompositions scaled.** a) All timescales, b) decadal, c) interannual, d) intra-annual.

represent the long-anticipated decline in lake phosphorus concentration in response to best management practices and remediation efforts of surface water sources.

The interannual components provide no clear covariance, while the intra-annual comparison suggests strongly phase-locked dynamics over most years. To explore cross variable dependence, we use CCM.

## Convergent cross mapping

Convergent cross mapping assesses the extent to which states of variable $X$ can be predicted from variable $Y$. If predictability using the entire time series (the full *library* of states) is significant, and, if predictability increases and converges as the state-space provides improved representations of the dynamics with increasing library size, it indicates shared dynamics and a causal link [31]. Fig 4 shows CCM results applied to milk production and phosphorus at different time scales.

We note that the nomenclature for cross mapping is X:Y, indicating that states of X are used to predict states of Y, and conversely, Y:X means that states of Y are used to predict states of X. Sugihara et al. [31] suggest that causality is inferred by measuring the extent to which the historical record of Y can reliably estimate states of X. This happens only if X is causally influencing Y. That is, if X is causally influencing Y, then states of X are "encoded" in states of Y. If states of Y are then able to predict states of X as dynamical information represents the complete dynamics (increasing library size) then one can infer that X is influencing Y. Causality is detected in the *upstream* direction. This means that links for Y causing X are denoted X:Y.

The top row of Fig 4 plots a) CCM with all time scales, and, b) with removal of the highest frequency IMF effectively implementing a low pass filter removing high frequency noise. Here, we find evidence that milk production can be considered a causal driver of lake phosphorus when all time scales are included in the system dynamics, with slightly clearer evidence when high frequency noise is removed.

Panel c) indicates no viable link between milk production and lake phosphorus on interannual time scales. One might expect a link between interannual climate conditions and milk

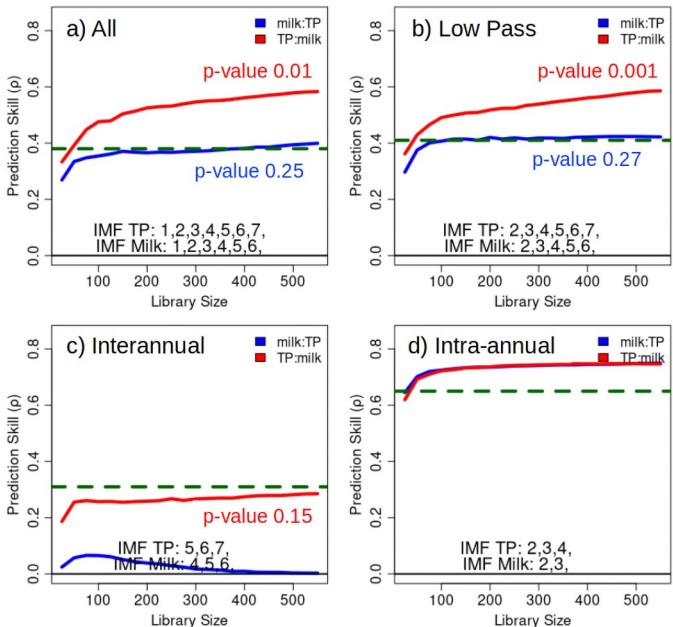

**Fig 4. Convergent cross mapping (CCM) of milk and lake phosphorus.** Dashed horizontal lines show linear cross correlation. a) All timescales, b) Low pass filter, c) interannual, d) intra-annual.

production since El-Niño conditions tend to produce cooler and wetter conditions in Florida [33], and, Holstien milk production increases as temperatures cool [34]. A more detailed investigation of this hypothesized link is shown in S1 Appendix El-Niño Southern Oscillation and milk production, providing additional details on the lack of an interannual link to milk production associated with ENSO.

The intra-annual, seasonal metric indicates strong coupling, but fails to resolve a directional attribute to the state predictions. This suggests a coupled, or common external driver on seasonal time scales. Seasonal maximums of lake phosphorus are associated with wind forcing during the winter [35], and, as noted above, milk production is also enhanced by winter temperature reduction.

## Conclusion

Decades of prosperity and attendant agricultural productivity have indelibly altered the land-use and ecosystems of central and south Florida. The addition of nutrient concentrations at order-of-magnitudes above pre-development levels have created a large reservoir of phosphorus in Lake Okeechobee. The waters of the lake are an important source of water supply for the Comprehensive Everglades Restoration Plan, but strict phosphorus limits must be met for this water to be admitted into the Everglades. Impressive phosphorus remediation efforts have been undertaken this Century, including restoration of natural flow paths to portions of the Kissimmee River [36], however, owing to the large amount of legacy phosphorus and complex dynamics of phosphorus monitoring, traditional data processing has not identified a decline of concentrations in the lake.

Using tools from nonlinear dynamical systems analysis, we find evidence of a reduction in mean phosphorus concentration over the last decade. Continued monitoring will reveal

whether this reflects the long-anticipated secular trend in phosphorus reduction, or, a temporary decline.

Additionally, we find that when data are viewed across all time scales, there is an apparent causal link between milk production and phosphorus concentration in the lake. This verifies the importance of continued remediation and source control efforts to mitigate phosphorus runoff.

## Supporting information

**S1 Appendix.**
(PDF)

## Author Contributions

**Conceptualization:** Joseph Park, Erik Stabenau.

**Data curation:** Joseph Park.

**Formal analysis:** Joseph Park, Erik Saberski.

**Funding acquisition:** Erik Stabenau, George Sugihara.

**Investigation:** Joseph Park.

**Project administration:** Joseph Park.

**Resources:** Erik Stabenau, George Sugihara.

**Supervision:** Erik Stabenau, George Sugihara.

**Visualization:** Erik Saberski.

**Writing – original draft:** Joseph Park.

**Writing – review & editing:** Joseph Park, Erik Saberski, Erik Stabenau, George Sugihara.

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
