## [Decision Letter · Decision Letter 0]

14 May 2021

PONE-D-21-07447

Dynamics of Florida milk production and total phosphate in Lake Okeechobee

PLOS ONE

Dear Dr. Park,

Thank you for submitting your manuscript to PLOS ONE. After careful consideration, we feel that it has merit but does not fully meet PLOS ONE’s publication criteria as it currently stands. Therefore, we invite you to submit a revised version of the manuscript that addresses the points raised during the review process.

I got the recommendations and comments from an expert reviewer on the field. The reviewer agreed that the data support the conclusions. However, lack of the explanation in Methods sections were suggested by the reviewer, and I totally share their comments. Especially, the article in its briefness fails to adequately aid the reader in following and interpreting their tautology and results. Therefore, I can invite you to submit a revised version of the manuscript that addresses the points raised by the reviewer.

We look forward to receiving your revised manuscript.

Kind regards,

Hideyuki Doi

Academic Editor

PLOS ONE

Journal Requirements:

2. Thank you for submitting the above manuscript to PLOS ONE. During our internal evaluation of the manuscript, we found significant text overlap between your submission and the following previously published works.

- https://www.nap.edu/read/25198/chapter/7#134

We would like to make you aware that copying extracts from previous publications, especially outside the methods section, word-for-word is unacceptable, even for works which you authored. In addition, the reproduction of text from published reports has implications for the copyright that may apply to the publications.

Please revise the manuscript to rephrase the duplicated text, cite your sources, and provide details as to how the current manuscript advances on previous work. Please note that further consideration is dependent on the submission of a manuscript that addresses these concerns about the overlap in text with published work.

4. We note that Figure 1 in your submission contains satellite images which may be copyrighted.

We require you to either (a) present written permission from the copyright holder to publish this figure specifically under the CC BY 4.0 license, or (b) remove the figure from your submission:

b. If you are unable to obtain permission from the original copyright holder to publish this figure under the CC BY 4.0 license or if the copyright holder’s requirements are incompatible with the CC BY 4.0 license, please either i) remove the figure or ii) supply a replacement figure that complies with the CC BY 4.0 license. Please check copyright information on all replacement figures and update the figure caption with source information. If applicable, please specify in the figure caption text when a figure is similar but not identical to the original image and is therefore for illustrative purposes only.

Additional Editor Comments:

I got the recommendations and comments from an expert reviewer on the field. The reviewer agreed that the data support the conclusions. However, lack of the explanation in Methods sections were suggested by the reviewer, and I totally share their comments. Especially, the article in its briefness fails to adequately aid the reader in following and interpreting their tautology and results. Therefore, I can invite you to submit a revised version of the manuscript that addresses the points raised by the reviewer.

Reviewers' comments:

Reviewer's Responses to Questions

**Comments to the Author**

1. Is the manuscript technically sound, and do the data support the conclusions?

Reviewer #1: Yes

2. Has the statistical analysis been performed appropriately and rigorously? 

Reviewer #1: Yes

3. Have the authors made all data underlying the findings in their manuscript fully available?

Reviewer #1: No

4. Is the manuscript presented in an intelligible fashion and written in standard English?

Reviewer #1: Yes

5. Review Comments to the Author

Reviewer #1: Overall:

I thought the manuscript provided a novel application to assessing the causality of drivers of phosphorus dynamics in Lake Okeechobee using the EMD, EDM, and CCM approaches. My main concern is the article in its briefness fails to adequately aid the reader in following and interpreting their tautology and results. While I am familiar with the basics of EMD, EDM, and CCM, the average reader will not be, especially those involved in the science and management of Lake Okeechobee. In this system in particular, it is critical that the results and the logic of the inference is clear to the reader. It is not in this manuscript.

Introduction overall:

It would be useful for the reader in the “Contemporary Conditions and Restoration” or “Dynamical Perspective” sections to make it abundantly clear to the reader that your concern is with the loading of phosphorus to the lake and not the in-lake phosphorus concentration. While they are invariably linked, the legacy phosphorus in Lake Okeechobee is likely to have a larger impact on downstream conditions than the in-flow conditions. In-lake total phosphorus has increased dramatically in recent years and around hurricane events. Minimal decreases in in-flow concentrations are likely to have almost no effect in the short-term on the outflow conditions given the lake’s size, circulation, and immense storage of legacy phosphorus.

Line 71-72: This is insufficient information to justify your choice of IMFs in subsequent analyses. The average reader will not be able to understand your methodological choices (lines 95-99) and their impacts without further information.

Line 76-77: This is again insufficient information for the average reader to understand your methodological choices (lines 95-99) and their impacts.

Materials and methods:

Overall: There is no mention of El Niño in the methods at all, just the results.

Line 84-87: This is insufficient information to recreate your search query. Provide more information. Did you use all milk production in Florida? Or particular counties in the Lake Okeechobee watershed? Justify your choice.

Line 88: Which stations? Siders and Havens (2020) shows and discusses how the DBHYDRO stations have changed in sampling scope over time. How do you assess whether these stations were reliable. Did all stations report across the whole time series or were they biased to particular inflows? In earlier years of the time series, the sampling was particularly sparse, how was this handled? Stations nearer to the Kissimmee River inflow are probably the only reliable stations to use to measure the inflow dynamics.

Line 91: How does this spline perform? What is the coverage of the original data? Given the changing sampling patterns of the DBHYDRO stations over time, how reliable is the spline at interpolating the information. Siders and Havens (2020) shows that the sampling is highly autocorrelated, how did you handle this in your interpolation.

Line 88-91: Why choose in-lake stations at all if the focus is on the inflow phosphorus dynamics?

Line 97: How did you determine they had interannual frequencies? If this is intrinsic to IMFs, then this should be clear earlier in the manuscript.

Results:

Line 131: There is only five decades in the data. I’m not sure how this is terribly surprising just given human population growth in the region. It seems that a remarkably high amount of time series would show the same trend when scaled.

Line 135: That data used was on in-lake concentrations not loadings, this statement is not accurate.

Line 146-149: The phrasing on the CCM notation is exceptionally confusing. Please rephrase so the reader can clearly understand the take away.

Line 151: The low-pass filter is not mentioned in the methods.

Line 156-159: This is the first mention of El Niño. It is confusing.

Line 160-162: Seasonal patterns in Lake Okeechobee have been discussed at length in the literature. This should be discussed.

Discussion:

Line 167: Citation?

Line 174-177: That data used was on in-lake concentrations not loadings, this statement is not accurate.

Figures:

Figures 2 and 3 can be merged.

Figures 3 and 4 should be color-blind friendly.

6. PLOS authors have the option to publish the peer review history of their article (what does this mean?). If published, this will include your full peer review and any attached files.

Reviewer #1: No

---

## [Author Response · Author response to Decision Letter 0]

1 Jul 2021

Please see the attached response to reviwers .pdf.

---

## [Editor Report · Decision Letter 1]

21 Jul 2021

Dynamics of Florida milk production and total phosphate in Lake Okeechobee

PONE-D-21-07447R1

Dear Dr. Park,

We’re pleased to inform you that your manuscript has been judged scientifically suitable for publication and will be formally accepted for publication once it meets all outstanding technical requirements.

Kind regards,

Hideyuki Doi

Academic Editor

PLOS ONE

Additional Editor Comments (optional):

I and the reviewer checked the revised manuscript as well as the response letter. The reviewer reported that most of my previous review were based on methodological issues that, if resolved, should permit the manuscript to be worthy of publication. So, I agree the revisions according to the reviewers’ comments and now can recommend to publish the paper in this journal.
---

## [Editor Report · Acceptance letter]

28 Jul 2021

PONE-D-21-07447R1 

Dynamics of Florida milk production and total phosphate in Lake Okeechobee 

Dear Dr. Park:

I'm pleased to inform you that your manuscript has been deemed suitable for publication in PLOS ONE. Congratulations! Your manuscript is now with our production department. 

Kind regards, 

on behalf of

Dr. Hideyuki Doi 

Academic Editor

PLOS ONE